# GABA Application Enhances Drought Stress Tolerance in Wheat Seedlings (*Triticum aestivum* L.)

**DOI:** 10.3390/plants12132495

**Published:** 2023-06-29

**Authors:** Qiuyan Zhao, Yan Ma, Xianqing Huang, Lianjun Song, Ning Li, Mingwu Qiao, Tiange Li, Dan Hai, Yongxia Cheng

**Affiliations:** Henan Engineering Technology Research Center of Food Processing and Circulation Safety Control, College of Food Science and Technology, Henan Agricultural University, Zhengzhou 450002, China

**Keywords:** GABA, drought stress, wheat seedlings, phenolic acids, antioxidant system

## Abstract

In this study, the effects of γ-aminobutyric acid (GABA) on physio-biochemical metabolism, phenolic acid accumulation, and antioxidant system enhancement in germinated wheat under drought stress was investigated. The results showed that exogenous GABA reduced the oxidative damage in wheat seedlings caused by drought stress and enhanced the content of phenolics, with 1.0 mM being the most effective concentration. Six phenolic acids were detected in bound form, including *p*-hydroxybenzoic acid, vanillic acid, syringic acid, *p*-coumaric acid, ferulic acid, and sinapic acid. However, only syringic acid and *p*-coumaric acid were found in free form. A total of 1.0 mM of GABA enhanced the content of total phenolic acids by 28% and 22%, respectively, compared with that of drought stress, on day four and day six of germination. The activities of phenylalanine ammonia lyase (PAL), cinnamic acid 4-hydroxylase (C4H) and 4-coumarate coenzyme A ligase (4CL) were activated by drought stress plus GABA treatment. Antioxidant enzyme activities were also induced. These results indicate that GABA treatment may be an effective way to relieve drought stress as it activates the antioxidant system of plants by inducing the accumulation of phenolics and the increase in antioxidant enzyme activity.

## 1. Introduction

Drought is considered an important abiotic stress with major impacts on plant growth and productivity [1]. Drought stress causes physiological, biochemical, and morphological changes in plants. In addition, respiration, photosynthesis, enzyme activities, redox homeostasis, and chloroplast metabolism are also affected by drought [2]. Plants have evolved different defense systems to avoid oxidative damage caused by drought stress, including overproduction of antioxidant metabolites that prevent the spread of the oxidative chain reaction. In this case, polyphenols such as phenolic acids, flavonoids, and anthocyanins play an important role in reducing the adverse effects of stress [3].

Wheat (*Triticum aestivum* L.), as one of the earliest cultivated gramineae plants in the world, is an important grain crop in many countries. Wheat grains are usually ground into flour to make food for satiation. In recent years, germinated grains have attracted much attention due to their unique function and nutritional value. Studies have shown that wheat sprouts are good sources of minerals (potassium, magnesium, and calcium), vitamins (vitamins C and E), enzymes (superoxide dismutase and catalase), amino acids, bioflavonoids, phenolic acids (ferulic acid and vanillic acid), etc. [4]. Among the bioactive phytochemicals, phenolics were cited as the most important contributors to the antioxidant capacity of cereal grains and have many physiological functions in the human body [5]. At present, most studies on germinated wheat focus on the regulation of germination conditions [6] and the change in the processing properties of malt flour [7]. In recent years, more and more studies have focused on the content and composition of phenolics in wheat germination from different countries during germination [8,9,10].

Phenolics are secondary metabolites of plants, which are mainly synthesized by the phenylpropane metabolic pathway, and phenylalanine ammonia lyase (PAL), cinnamic acid 4-hydroxylase (C4H), and 4-coumarate CoA ligase (4CL) are the key enzymes involved in its biosynthesis. The biosynthesis of phenolics is a complex network of chemical reactions and an endogenously regulated process during plant growth and development [11], or which can be stimulated by exogenous factors such as moisture [12,13]. Environmental stresses induce the accumulation of phenolic compounds to protect tissues from damage caused by free-radical-induced oxidative stress [14]. When wheat and other grains germinated under stress conditions such as drought, stress signals were recognized and amplified by cell membrane receptors, related endogenous enzymes were activated, and the metabolic direction of substances changed. Finally, the composition and content of phenolics changed [15]. At present, studies on the key enzymes of phenolic biosynthesis in wheat seedlings mostly focus on changes in PAL enzyme activity [8,16]. There are few studies on C4H and 4CL during wheat germination under stress, and the study on the relationship between phenolics and the key enzymes of phenolic biosynthesis during wheat germination under stress needs to be strengthened.

γ-Aminobutyric acid (GABA) is a four-carbon non-protein amino acid and is widespread in nature. Previous studies have shown that GABA accumulated rapidly in various stresses, which implied that GABA might play a dual role as both a metabolite and a legitimate plant-signaling molecule [17,18]. The application of GABA under different abiotic stresses has been proven to alleviate stress by regulating the metabolism of sugar and proline [19,20], alleviating the effects of photosynthesis and mitochondrial activity inhibition and maintaining the integrity of chloroplasts under stress [19,20], increasing the activity of guaiacol peroxidase, catalase, superoxide dismutase, ascorbate peroxidase, and other antioxidant enzymes [19,20], decreasing Na^+^ concentration and increasing K^+^ concentration [21], increasing amino acid and organic acid accumulation [22], promoting the production of polyamines (PAs) and the inhibition of their metabolism [23], and increasing the content of unsaturated fatty acids (γ -linolenic acid) and reducing the content of saturated fatty acids [24]. In wheat, studies have shown that GABA increased salt tolerance by promoting nitrogen and carbon assimilation [25], improving photosynthesis, and enhancing the activities of antioxidant enzymes [26], as well as altering signal transduction pathways [27]. While under drought stress, GABA reduced the activity loss or degeneration of some enzymes and proteins, so as to maintain the balance of cell metabolism [28]. The mechanism of GABA is different for plants to resist different stresses, such as salt stress and drought stress. Additionally, studies have shown that GABA may respond to stress by inducing phenolic accumulation and enhancing antioxidant activity in barley seedlings [29] and soybean sprouts [30] under NaCl stress. To our knowledge, the regulatory role of GABA in the biosynthesis of phenolics induced by drought stress in germinated wheat has not been reported in the literature.

In this study, the effects of GABA on phenolic acids and antioxidant systems in wheat seedlings under drought-stress conditions were investigated. This would provide the basis for further research on the mechanism of GABA in enhancing plant stress resistance and bring new ideas for the enrichment of phenolics.

## 2. Results 

### 2.1. Effects of GABA on Length and Weight of Wheat Seedlings under Drought Stress

Changes in the length and weight of wheat seedlings under different treatments were presented in Figure 1. With the prolonging of germination time, the length of wheat seedlings increased significantly (Figure 1A). The response of seedling fresh weight to different germination stages was similar to that of seedling length, and it increased significantly during germination (Figure 1C). However, a slight decrease in freeze-dried weight was observed (Figure 1D). Drought stress resulted in a significant reduction in seedling length during the whole growth process (Figure 1A), but root length only decreased at the initial stage of stress and had no significant change at the later stage (Figure 1B). The fresh weight of wheat seedlings also declined markedly under drought stress (Figure 1C). Meanwhile, PEG6000 treatment caused a considerable increase in freeze-dried weight (Figure 1D). Exogenous GABA treatment significantly alleviated the growth inhibition of wheat seedlings under drought stress, and the seedling length were enhanced by 2.69%, 19.53%, 34.27%, 19.67% and 16.68% on the 6th day, respectively.

### 2.2. Effects of GABA on the MDA Content and Electrolyte Leakage of Wheat Seedlings under Drought Stress

As shown in Figure 2, germination resulted in a significant increase in the MDA level and electrolyte leakage. The largest increase in MDA level occurred from 4 d to 6 d of germination, and the largest increase in electrolyte leakage occurred from 2 d to 4 d of germination. Drought stress led to pathophysiological stress and damage of germinated wheat, which caused a noticeable improvement in MDA level and electrolyte leakage of 83% and 33%, respectively, on day 6 of germination, compared with the control. Application of exogenous GABA significantly alleviated the growth-inhibiting effects of drought stress. The maximum reduction in electrolyte leakage and MDA level of wheat treated with drought stress plus 1.0 mM GABA was 23% and 26%, respectively, after germination for 6 d, compared with that of drought stress.

### 2.3. Effects of GABA on the Total Phenolic Content of Wheat Seedlings under Drought Stress 

Results showed that the total phenolic content on day 4 and day 6 of germination induced by drought stress increased by 64% and 29%, respectively (Figure 3). Exogenous GABA significantly increased total phenolic content under drought stress, which were 1.05, 1.10, 1.21, 1.15, and 1.11 times higher than that under only drought stress, respectively, under 0.25 mM, 0.5 mM, 1.0 mM, 2.0 mM, and 4.0 mM GABA at the 6 d of germination. The addition of 1.0 mM GABA had the most significant effect, resulting in an increase in the total phenolic content of 116% and 121% relative to that of drought stress on day 4 and day 6 of germination.

### 2.4. Effects of GABA the on Phenolic Acids Content of Wheat Seedlings under Drought Stress 

As can be seen from Figure 4, phenolic acids are mainly in the bonded form in germinated wheat. Six phenolic acids were found in combined form, including *p*-hydroxybenzoic acid, vanillic acid, syringic acid, *p*-coumaric acid, ferulic acid, and sinapic acid. However, only syringic acid and *p*-coumaric acid were found in free form. 

Ferulic acid was the main phenolic acid in wheat seedlings. Drought stress induced a significant increase of 30% and 35% of bound ferulic acid on day 4 and day 6 of germination, respectively, compared with that of the control. A comparison of the 100 g/L PEG 6000 plus 1.0 mM GABA treatment with the drought stress on wheat seedlings showed that the bound ferulic acid content significantly increased by 26% and 39%, compared with drought stress on day 4 and day 6 of germination, respectively. Ferulic acid content also increased by 26% and 25% on day 6 of germination under 0.5 mM and 2.0 mM GABA treatment, but there was no significant difference under 0.25 mM and 5.0 mM GABA treatment compared with that of drought stress. 

*p*-Coumaric acid existed in both free and bound forms in wheat seedlings, but the bound form is the main state of existence. Under drought stress, the content of free *p*-coumaric acid did not increase significantly, but the content of bound *p*-coumaric acid was 1.23 and 1.27 times higher than the control on day 4 and day 6 of germination. The content of *p*-coumaric acid varied with different concentrations of GABA at different germination stages. The exogenous application of 1.0 mM GABA induced a significant increase of 38% of total *p*-coumaric acid on day 4 of germination, but led to a decrease of 26% on day 6 of germination, compared with that under drought stress.

Sinapic acid was only detected in bound form. Exogenous application of 1.0 mM GABA under drought stress significantly increased the sinapic acid content, which were 1.23 and 1.24 times higher than that under only drought stress, respectively, on day 4 and day 6 of germination. Other GABA concentrations also had different effects on sinapic acid content, which increased significantly at concentrations of 0.5 mM, 1.0 mM, and 2.0 mM in the late stage of germination, but there was no significant difference under 0.25 mM and 5.0 mM GABA treatment, compared with that of drought stress.

Other phenolic acids in wheat seedlings had a lower content. *p*-Hydroxybenzoic acid and vanillic acid were only detected in bound form, and syringic acid was found both in free and bound form. *p*-Hydroxybenzoic acid was not found in the control treatment at the early stage of germination until day 6. However, *p*-Hydroxybenzoic acid was detected both under drought stress and PEG plus GABA treatment, and its content decreased significantly compared with the control. The content of vanillic acid was significantly increased at different concentrations of GABA treatment in the initial stage of germination, but decreased at the later stage compared with that of drought stress. Syringic acid increased significantly when treated with exogenous GABA. The content of the bound form was significantly increased at the early stage of germination, and the free form was significantly increased at the late stage of germination.

### 2.5. Effects of GABA on the PAL, C4H, and 4CL Activity of Wheat Seedlings under Drought Stress 

As shown in Figure 5, the activities of PAL, C4H, and 4CL enzymes increased significantly at the early stage of germination. However, the activities of PAL and 4CL decreased at the late stage of germination (Figure 5). The activities of PAL, C4H, and 4CL in wheat seedlings under drought stress were 1.25, 1.30, and 1.23 times, and 1.34, 1.29, and 1.35 times higher than those of the control, respectively, on day 4 and day 6 of germination. The addition of GABA significantly enhanced the activities of PAL, which increased by 35%, 51%, 63%, 27%, and 11% compared with that of drought stress, respectively, under 0.25 mM, 0.5 mM, 1.0 mM, 2.0 mM, and 4.0 mM GABA at the 6 d of germination. GABA treatment also significantly increased the activities of C4H and 4CL, which were 1.40 and 1.94 times higher than those of the control, respectively, on day 6 of germination.

### 2.6. Effects of GABA on the POD, CAT and APX Activity of Wheat Seedlings under Drought Stress 

As shown in Figure 6, the activity of the antioxidant enzymes of POD, CAT, and APX was determined. Higher levels of antioxidant enzyme activity were observed under drought stress. Compared with drought stress, exogenous GABA increased the activities of POD and CAT during the whole experiment. Under NaCl + 0.1 mM GABA treatment, APX activity increased significantly on day 4, but there was no significant difference on day 6 when compared with that of drought stress.

### 2.7. Effects of GABA on the Antioxidant Capacity of Wheat Seedlings under Drought Stress

As shown in Figure 7, the antioxidant capacity of wheat seedlings under different treatments was expressed by ABTS and DPPH radical scavenging capacity. The radical scavenging ability of wheat seedlings was enhanced under drought stress, which was similar to the change in total phenolic content. There was a concentration gradient effect of GABA on free radical scavenging activity. Exogenous GABA enhanced the radical scavenging activity of wheat seedlings, with 1.0 mM being the most effective concentration. The 0.1 mM GABA treatment increased the ABTS and DPPH values by 24% and 28% on day 4, and 30% and 34% on day 6 of germination, compared with those of the wheat seedlings under drought stress alone. 

## 3. Discussion

Drought stress is one of the most serious problems for plants all over the world. It has a negative effect on plants because the available water in the root zone is lower than the water required to maintain optimal growth and development [31]. In the present study, drought stress caused a noticeable increase in MDA levels and electrolyte leakage (Figure 2), indicating that drought stress caused osmotic stress. The increased accumulation of EL and MDA was considered to be an indicator of membrane damage and lipid peroxidation caused by excessive ROS production during stress [23]. As a result, plant dehydration and water uptake were inhibited, and the concentration of cellular components increased to levels that led to membrane damage and even cell death [12,31], which resulted in significant reductions in seedling length and fresh weight throughout growth (Figure 1). The application of exogenous GABA alleviated the growth stress caused by drought stress effectively (Figure 1). This was consistent with the results of Abd El-Gawad et al., in which exogenous GABA increased the shoot weight and leaf area of *Phaseolus vulgaris* L [32]. The positive effects of the exogenous application of GABA on mitigating drought stress has been demonstrated in a variety of plant species from different perspectives. Among them, improving osmotic regulation and maintaining membrane stability were recognized as protective effects [19,20,32]. Numerous studies have explored the role of GABA in wheat seedlings under abiotic stress, and there are many similarities between salt stress and drought stress. For example, GABA could resist the damage caused by salt or drought stress by promoting the production of proline, soluble sugar, and other osmotic regulating substances in wheat seedlings [20,27]. The difference is that GABA also reduced the activity loss or degeneration of some enzymes and proteins under drought stress, so as to maintain the balance of cell metabolism [28]. Under salt stress, GABA not only made the osmotic regulation system and antioxidant system respond, but also changed the signal transduction pathway [27]. Studies showed that GABA alleviated drought stress with a concentration gradient effect, in which the electrolyte leakage and MDA content of wheat seedlings increased more slowly under 1.0 mM of GABA treatment. The results showed that GABA could protect wheat seedlings from oxidative damage and thus improve drought resistance.

It is well known that drought stress triggers enzymatic and non-enzymatic antioxidant defenses leading to redox homeostasis. The proven antioxidant activity of phenolics enables them to act as active oxygen scavengers. Therefore, their synthesis is usually triggered by biotic/abiotic stresses, especially under salt and drought stress [33]. In the present study, our results depicted that drought stress resulted in a marked increase in total phenolic (Figure 3) and phenolic acid content (Figure 4). Interestingly, our study showed that the increase in the variety and content of bound phenolic acids was a major contributor to the increase in phenolic acids. Phenolic acids mainly existed in their bonded form in germinated wheat. Six phenolic acids were found in combined form, but only syringic acid and *p*-coumaric acid were found in free form. This may have something to do with the type of wheat we chose. It was generally believed that the composition of phenolic acids of wheat seedlings varied dynamically under different varieties and culture conditions, but the main phenolic acids such as ferulic acid and *p*-coumaric acid tended to increase with increasing germination time [8,9,10]. By the same token, we achieved the same result. The accumulation of phenolic acids in response to drought stress was mainly caused by the increase in two major compounds, ferulic acid and *p*-coumaric acid. Drought stress can damage the photosynthetic organs of plants through the interaction with ultraviolet or visible light, while ferulic acid can transform the light falling on the leaves into blue fluorescence to make it no longer destructive and provide protection for photosynthesis [34]. In addition, a new phenolic acid, *p*-hydroxybenzoic acid, emerged under drought stress, which was not detected in the early stage of wheat seed germination. Similar results were found for other plant species, where drought also increased the content and diversity of phenolic compounds [12,35,36]. 

The addition of GABA significantly improved the activity of PAL, C4H, and 4CL and displayed a higher total phenolic (Figure 5) and phenolic acids content (Figure 4) than that of drought stress, which was consistent with previous studies [29,30]. The increase in total phenolic and phenolic acid content was attributed to the enhanced activity of key enzymes involved in phenolic synthesis. PAL was involved in the first step of the phenylpropionate pathway and has been suggested to act as a rate-limiting enzyme regulating the overall flux into phenylpropionate metabolism [37]. The change in PAL activity was consistent with the content of total phenolic at the initial stage of germination. However, on the 6th day of germination, the activity decreased, but the total phenolic content was still increasing, which may be related to the cumulative effect of phenolic compounds. The cytochrome P450 monooxygenase, cinnamate 4-hydroxylase (C4H), is the second major gene in the phenylpropanoid pathway branch [37]. This enzyme catalyzes the hydroxylation of trans-cinnamic acid to *p*-coumaric acid, which provides the feedstock for eventual conversion to various phenolic acids, including ferulic acid and vanillic acid [36,37]. In fact, our study showed a high correlation between C4H and the total phenolic and total phenolic acid (r = 0.927 or 0.923, *p <* 0.01) (Appendix A), and the activity of C4H was also highly positively correlated with the accumulation of major phenolic acids such as ferulic acid, *p*-coumaric acid, and sinapic acid (r = 0.938, 0.592, and 0.802, respectively, *p <* 0.01 or *p <* 0.05) (Appendix A). These results help to confirm that C4H activity is closely related to the accumulation of phenolic acids. 

Plants respond to drought stress by increasing their antioxidant defenses, which involve several compounds, such as anthocyanins, flavonoids, and phenolic acids. In addition, antioxidant enzymes are involved, such as peroxidase (POD), catalase (CAT), ascorbate peroxidase (APX), etc. Ascorbate is a ubiquitous non-enzymatic antioxidant that not only has significant potential to scavenge reactive oxygen species, but can also affect some key processes in plants under both stressed and non-stressed conditions [37,38]. In this study, drought stress caused a significant increase in the POD and APX enzyme activities (Figure 6), suggesting that they played an integrative role in regulating drought resistance in wheat seedlings. However, drought stress resulted in a marked decrease in the CAT enzyme activity, which was possibly due to the inactivation of the enzyme by overproduction of reactive oxygen species or due to interference with the synthesis of the enzyme or its subunits [39]. Similar findings were observed in sunflowers (*Helianthus annuus* L.) subjected to drought and heat stress [40]. Exogenous application of GABA improved the activities of POD, CAT, and APX enzymes when subjected to drought stress. Previous reports have confirmed the role of GABA in the regulation of antioxidant enzyme defense in other plants [38,40], suggesting that GABA plays a key role as a mitigating agent against drought-induced oxidative stress and the up-regulation of defense responses.

The antioxidant capacity of germinated wheat was determined in vitro using two different antioxidant assays (ABTS and DPPH) (Figure 7). Both the ABTS and DPPH radical scavenging activity of wheat seedlings increased in response to drought stress, which was consistent with the change in the antioxidant capacity of Amaranthus leafy vegetable [35]. Exogenous GABA enhanced the radical scavenging activity of wheat seedlings, with 1.0 mM being the most effective concentration. It has been proved that the increase in antioxidant capacity is due to the enrichment of phenolic compounds [41]. Pearson’s correlation coefficient revealed that antioxidant capacity measured by ABTS and DPPH was strongly correlated with syringic acid, *p*-coumaric acid, ferulic acid, and sinapic acid, which means that these phenolic acids were the highest contributors to the antioxidant activity determined by ABTS and DPPH (Appendix A). The results showed that GABA was involved in the enhancement of antioxidant systems in wheat seedlings. 

## 4. Materials and Methods

### 4.1. Material and Experimental Design

The wheat variety Qiule 168 was used as the research object. “Qiule 168” is a wheat variety independently cultivated by Henan Qiule Seed Industry Science and Technology Co., Ltd. using the genealogy method, and approved by Henan Provincial Clothing Crop Variety Examination Committee in 2019 (Yumai 20190008). “Qiule 168” is a hybrid bred from “Yumai 34” as female parent and “Zhoumai 13” as male parent.

Wheat seeds were soaked in 0.5% sodium hypochlorite for 15 min and then washed. The seeds were then immersed in deionized water at 25 °C for 6 h at a ratio of 1:5 (*w*/*v*). All seeds were then evenly scattered on an automatic spray system tray and placed in a growing chamber at 25 °C. The seeds were treated with deionized water for 2 days first. Then, different treatments were carried out from the 3rd day as follows: (1) CK: the wheat was sprayed with distilled water. (2) PEG: the wheat was sprayed with 100 g/L PEG 6000. (3) PEG + GABA1: 100 g/L PEG 6000 and 0.25 mM GABA were sprayed on the wheat. (4) PEG + GABA2: 100 g/L PEG 6000 and 0.5 mM GABA were sprayed on the wheat. (5) PEG + GABA3: 100 g/L PEG 6000 and 1.0 mM GABA were sprayed on the wheat. (6) PEG + GABA4: 100 g/L PEG 6000 and 2.0 mM GABA were sprayed on the wheat. (7) PEG + GABA5: 100 g/L PEG 6000 and 4.0 mM GABA were sprayed on the wheat. The wheat seeds germinated for a total of 6 days.

### 4.2. Measurement of Length and Weight 

The wheat seedlings′ lengths were measured with vernier calipers, and twenty wheat seedlings were selected and measured. The result was expressed in mm/seedlings.

The weight was determined by randomly selecting 100 wheat seedlings as the sampling group, a total of 3 groups were selected to take the average value, and the results were expressed as g/100 seedlings. For the weight of fresh wheat seedlings, 100 fresh wheat seedlings were weighed in an analytical balance. For the weight of dry wheat seedlings, 100 fresh wheat seedlings were dried in an electrically heated air-drying oven and then cooled in a dryer.

### 4.3. Determination of Malondialdehyde (MDA) Content and Electrolyte Leakage

MDA content was determined by the method of Hodges [42]. Briefly, fresh wheat seedlings (1 g) were mashed and transferred to a solution of 10% trichloroacetic acid (5 mL). After centrifugation at 10,000× *g* for 20 min at 4 °C, the supernatant was mixed with 0.67% thiobarbituric acid, and the mixture was immersed in boiling water for 30 min. After cooling, the supernatant was centrifuged again. The absorbance of the supernatant was measured at 450, 532, and 600 nm. MDA content was calculated using the following equation:$$\mathrm{MDA}\;\mathrm{concentration}\;\left(\mu \mathrm{mol}/L\right)=6.45\times \left(A532-A600\right)-0.56\times A450\;\mathrm{MDA}\;\mathrm{concentration}\;(\mu \mathrm{mol}/g\;\mathrm{FW})=\frac{c\;\times \;5}{1}$$

Electrolyte leakage was measured using a conductivity meter. Fresh wheat seedlings were cut to 3 mm, placed in a tube and mixed with water. After oscillating for l h at 25 °C, the measured conductivity denoted EC1. The tube was then placed in a boiling water bath. After cooling, the measured boiling conductivity denoted EC2. The electrolyte leakage (EL) was calculated using the equation:EL (%) = EC1/EC2 × 100

### 4.4. Extraction of Phenolic Compounds

Free and bound phenolic compounds were extracted based on the methods reported by Ma et al. [29]. The dried extracts were redissolved with 50% methanol to a volume of 10 mL and used as crude free phenolic extracts. The extracts were stored at −20 °C for subsequent use.

### 4.5. Determination of Total Phenolic Content

The total phenolic content was determined using the Folin–Ciocalteu colorimetric method [29]. Methanol was used as the blank, and gallic acid (GA) was used as the standard. The total phenolic content was expressed as gallic acid equivalents (mg GAE/100 g DW, dry weight). 

### 4.6. Determination of Free and Bound Phenolic Acid Content

Free and bound phenolic compounds were filtered through a 0.45 μm membrane filter for a Shimadzu LC-2030 high-performance liquid chromatography (HPLC) system with a Zorbax SB-C18 column (5 µm particle size, 250 mm × 4.6 mm, Agilent Technologies Inc., Palo Alto, CA, USA) and a Variable Wavelength Detector (VWD). The mobile phase A was 1% acetic acid in water and the mobile phase B was methanol. The allowed time of separation of phenolic acids was 45 min at a constant temperature of 30 °C with 0.8 mL/min flow rate. A 45 min gradient was programmed as follows: 0–22 min, 18–28% B; 22–40 min, 28–40% B; 40–45 min, 40–18% B. 

### 4.7. Determination of Phenylpropanoid Metabolism-Related Enzymes Activity 

The PAL activity was determined as described by Assis et al. [43]. One unit of PAL activity was equal to a change of 0.01 at 290 nm per min, and expressed as U/g FW, dry weight.

Cinnamate-4-hydroxylase (C4H) activity was assayed following the method of Lamb and Rubery [44] with some modification. The 0.5 g of fresh tissue was blended with 5 mL of ice-cold 100 mM phosphate buffer (pH 7.6, containing 2 mM β-mercaptoethanol, 0.5 mM EDTA, 0.25 M sucrose) and ground at 4 °C. After centrifugation (12,000× *g*, 4 °C, 30 min), the supernatant was used for the enzyme assay. The reaction mixture contained 0.2 mL of 50 mM trans-cinnamic acid, 2 mL of 0.5 g/L NADPNa_2_, 3 mL of 100 mM phosphate buffer, and 0.1 mL crude enzyme. One unit of C4H activity was equal to a change of 0.01 at 340 nm per min, and expressed as U/g FW.

The activity of 4-coumarate coenzyme A ligase (4CL) was assayed based on Han et al. [45]. The absorbance was measured at 333 nm. One unit of 4CL activity was equal to a change of 0.01 in absorbance per min, and expressed as U/g FW.

### 4.8. Determination of Antioxidant Enzyme Activity

The activities of peroxidase (POD), catalase (CAT), and ascorbate peroxidase (APX) were measured according to the method reported previously [9]. One unit (U) of POD activity was defined as a change of 1 at 470 nm per minute and the results were expressed as U/g FW. One unit of CAT activity was defined as a change of 0.01 at 240 nm per minute and the results were expressed as U/g FW, and one unit of APX activity was defined as the change of 0.01 at 290 nm per minute and the results were expressed as U/g FW. 

### 4.9. Determination of Antioxidant Capacity 

The ABTS and DPPH radical scavenging activity assay was determined following the method of Chen et al. [9]. A plot of Trolox concentration with ABTS and DPPH radical scavenging rate was used as the standard curve. They were expressed as micromoles of Trolox equivalent (TE) per gram of sample (μmol TE/g DW).

### 4.10. Statistical Analysis

The results were presented as mean ± standard deviation (SD) of duplicated determination. The data were analyzed by one-way ANOVA using SPSS 18.0 (SPSS Inc., Chicago, IL, USA). The significance level was defined at *p* < 0.05 between treatments at the same germination time. 

## 5. Conclusions

GABA induced the accumulation of phenolic compounds, especially ferulic acid, *p*-coumaric acid, and sinapic acid, by regulating the activities of PAL, C4H, and 4CL involved in phenylpropanoid metabolism, with 1.0 mM being the most effective concentration. A total of 1.0 mM of GABA enhanced the content of total phenolic acids by 28% and 22%, respectively, compared with that of 100 g/L PEG 6000 treatment, on day 4 and day 6 of germination. Additionally, the activities of POD, CAT, and APX were induced in wheat seedlings. Pearson’s correlation coefficient revealed that antioxidant capacity was strongly correlated with syringic acid, *p*-coumaric acid, ferulic acid, and sinapic acid. The results showed that GABA was involved in the drought resistance of wheat seedlings through enhancing antioxidant capacity, which was achieved by inducing the accumulation of phenolic compounds and increasing the level of antioxidant enzyme activity.

## Figures and Tables

**Figure 1 plants-12-02495-f001:**
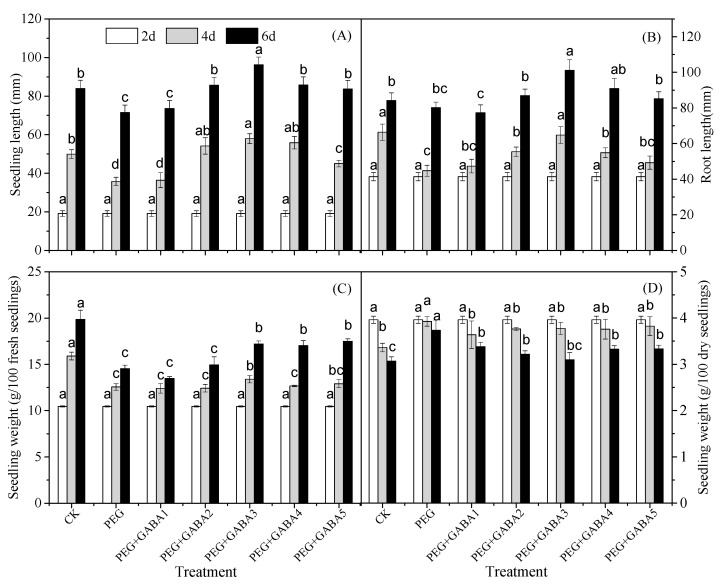
Effects of GABA on length and weight of wheat seedlings under drought stress. (**A**) seedling length, (**B**) root length, (**C**) fresh weight of wheat seedlings, (**D**) dry weight of wheat seedlings. The length and weight were determined on day 2, 4, and 6. Twenty wheat seedlings were selected and measured 20 times, and the average value was taken as the length of wheat seedlings and roots. One hundred wheat seedlings were selected as the sampling group, and a total of three groups were selected, whose average value was taken as the fresh weight and dry weight of wheat seedlings. Different letters following values indicate significant differences between treatments at the same germination time (*p* < 0.05). CK means the wheat was sprayed with distilled water; PEG means 100 g/L PEG 6000 treatment; PEG + GABA1 means 100 g/L PEG 6000 + 0.25 mM GABA treatment; PEG + GABA2 means 100 g/L PEG 6000 + 0.5 mM GABA treatment; PEG + GABA3 means 100 g/L PEG 6000 + 1.0 mM GABA treatment; PEG + GABA4 means 100 g/L PEG 6000 + 2.0 mM GABA treatment; PEG + GABA5 means 100 g/L PEG 6000 + 4.0 mM GABA treatment.

**Figure 2 plants-12-02495-f002:**
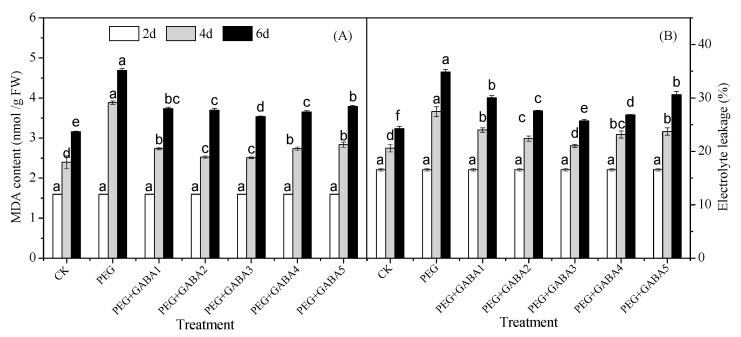
Effects of GABA on MDA content (**A**) and the electrolyte leakage (**B**) of wheat seedlings under drought stress. Different letters following values indicate significant differences between treatments at the same germination time (*p* < 0.05). CK, PEG, PEG+GABA1, PEG+GABA2, PEG+GABA3, PEG+GABA4, and PEG+GABA5 indicate distilled water treatment, 100 g/L PEG 6000 treatment, 100 g/L PEG 6000 + 0.25 mM GABA treatment, 100 g/L PEG 6000 + 0.5 mM GABA treatment, 100 g/L PEG 6000 + 1.0 mM GABA treatment, 100 g/L PEG 6000 + 2.0 mM GABA treatment, and 100 g/L PEG 6000 + 4.0 mM GABA treatment, respectively.

**Figure 3 plants-12-02495-f003:**
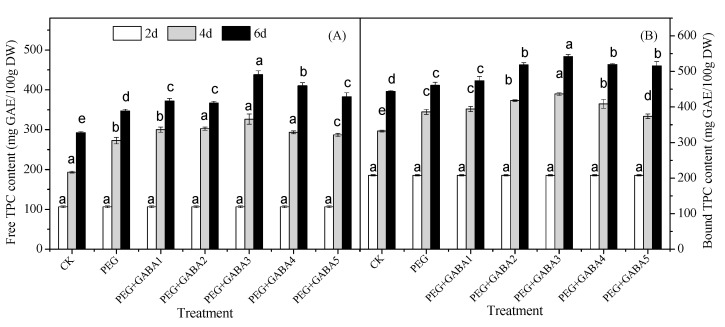
Effects of GABA on the free (**A**) and bound (**B**) total phenolic content of wheat seedlings under drought stress. Different letters following values indicate significant differences between treatments at the same germination time (*p* < 0.05). CK, PEG, PEG+GABA1, PEG+GABA2, PEG+GABA3, PEG+GABA4, and PEG+GABA5 indicate distilled water treatment, 100 g/L PEG 6000 treatment, 100 g/L PEG 6000 + 0.25 mM GABA treatment, 100 g/L PEG 6000 + 0.5 mM GABA treatment, 100 g/L PEG 6000 + 1.0 mM GABA treatment, 100 g/L PEG 6000 + 2.0 mM GABA treatment, and 100 g/L PEG 6000 + 4.0 mM GABA treatment, respectively.

**Figure 4 plants-12-02495-f004:**
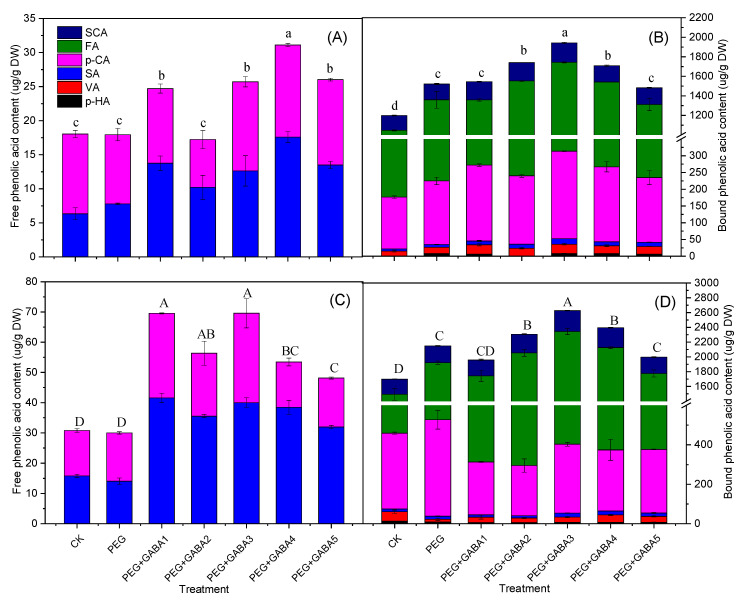
Effects of GABA on phenolic acids in wheat seedlings under drought stress: (**A**) free phenolic acid content of wheat seedlings on day 4, (**B**) bound phenolic acid content of wheat seedlings on day 4, (**C**) free phenolic acid content of wheat seedlings on day 6, (**D**) bound phenolic acid content of wheat seedlings on day 6. Values with different letters are significantly different at *p* < 0.05. Lowercase letters represent significant differences among treatment factors on day 4, and capital letters represent significant differences among treatment factors on day 6 of germination. CK, PEG, PEG+GABA1, PEG+GABA2, PEG+GABA3, PEG+GABA4, and PEG+GABA5 indicate distilled water treatment, 100 g/L PEG 6000 treatment, 100 g/L PEG 6000 + 0.25 mM GABA treatment, 100 g/L PEG 6000 + 0.5 mM GABA treatment, 100 g/L PEG 6000 + 1.0 mM GABA treatment, 100 g/L PEG 6000 + 2.0 mM GABA treatment, and 100 g/L PEG 6000 + 4.0 mM GABA treatment, respectively.

**Figure 5 plants-12-02495-f005:**
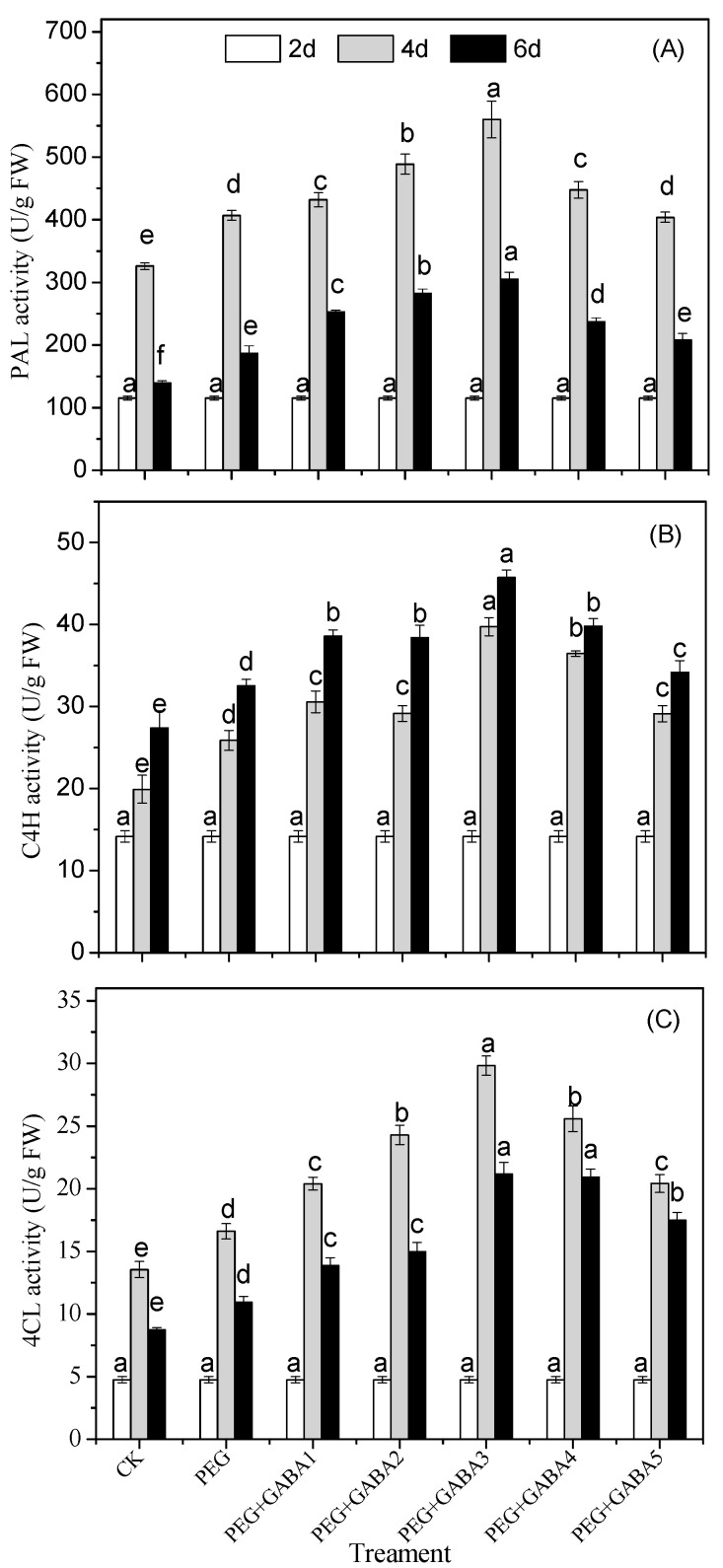
Effects of GABA on the PAL (**A**), C4H (**B**) and 4CL (**C**) activity of wheat seedlings under drought stress. Different letters following values indicate significant differences between treatments at the same germination time (*p* < 0.05). CK, PEG, PEG+GABA1, PEG+GABA2, PEG+GABA3, PEG+GABA4, and PEG+GABA5 indicate distilled water treatment, 100 g/L PEG 6000 treatment, 100 g/L PEG 6000 + 0.25 mM GABA treatment, 100 g/L PEG 6000 + 0.5 mM GABA treatment, 100 g/L PEG 6000 + 1.0 mM GABA treatment, 100 g/L PEG 6000 + 2.0 mM GABA treatment, and 100 g/L PEG 6000 + 4.0 mM GABA treatment, respectively.

**Figure 6 plants-12-02495-f006:**
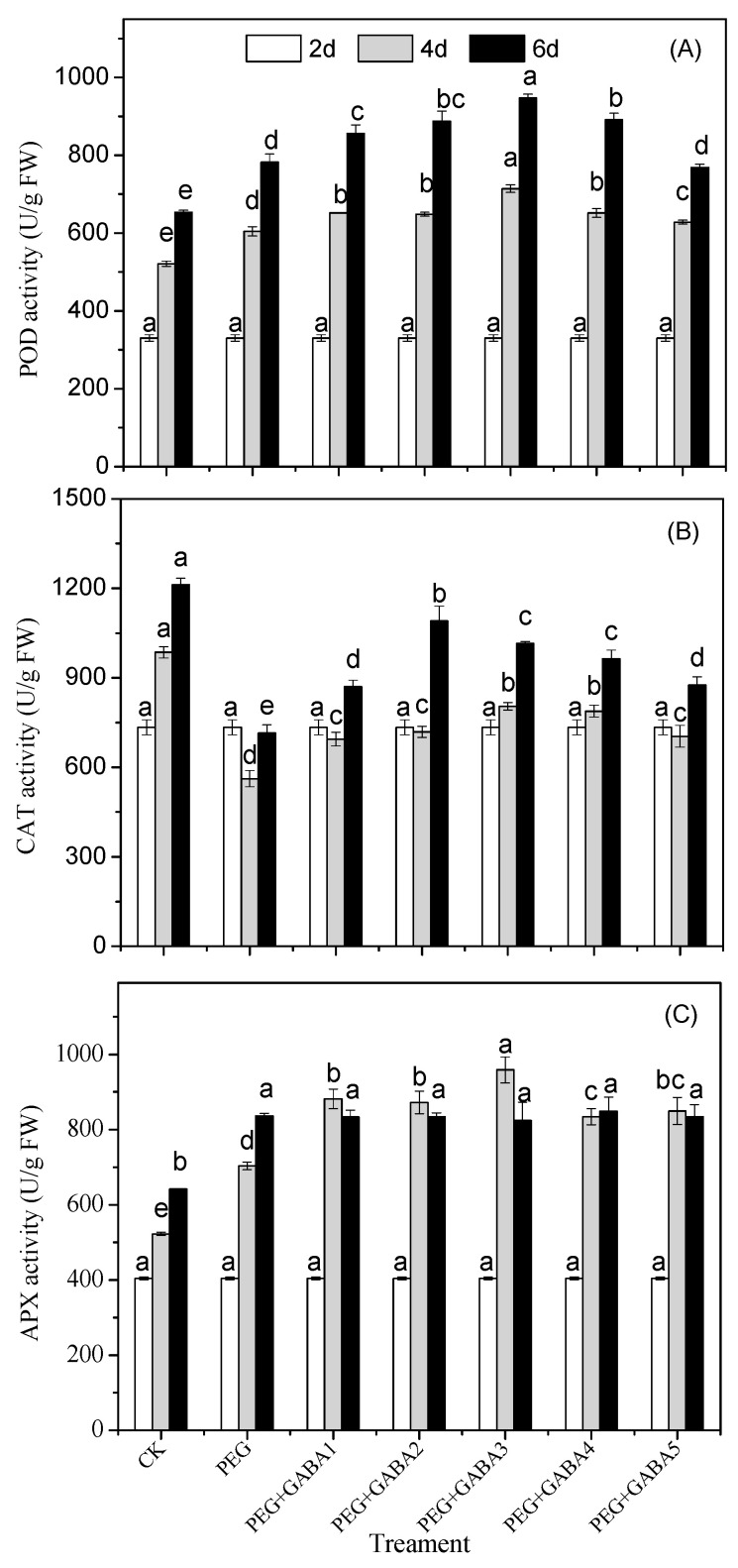
Effects of GABA on the POD (**A**), CAT (**B**) and APX (**C**) activity of wheat seedlings under drought stress. Different letters following values indicate significant differences between treatments at the same germination time (*p* < 0.05). CK, PEG, PEG+GABA1, PEG+GABA2, PEG+GABA3, PEG+GABA4, and PEG+GABA5 indicate distilled water treatment, 100 g/L PEG 6000 treatment, 100 g/L PEG 6000 + 0.25 mM GABA treatment, 100 g/L PEG 6000 + 0.5 mM GABA treatment, 100 g/L PEG 6000 + 1.0 mM GABA treatment, 100 g/L PEG 6000 + 2.0 mM GABA treatment, and 100 g/L PEG 6000 + 4.0 mM GABA treatment, respectively.

**Figure 7 plants-12-02495-f007:**
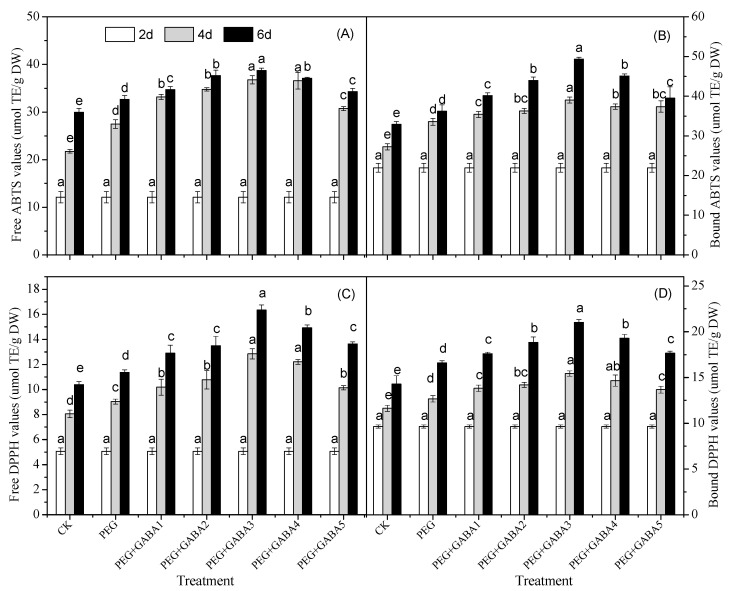
Effects of GABA on the ABTS values (**A**,**B**) and DPPH values (**C**,**D**) of wheat seedlings under drought stress. Different letters following values indicate significant differences between treatments at the same germination time (*p* < 0.05). CK, PEG, PEG+GABA1, PEG+GABA2, PEG+GABA3, PEG+GABA4, and PEG+GABA5 indicate distilled water treatment, 100 g/L PEG 6000 treatment, 100 g/L PEG 6000 + 0.25 mM GABA treatment, 100 g/L PEG 6000 + 0.5 mM GABA treatment, 100 g/L PEG 6000 + 1.0 mM GABA treatment, 100 g/L PEG 6000 + 2.0 mM GABA treatment, and 100 g/L PEG 6000 + 4.0 mM GABA treatment, respectively.

## Data Availability

Not applicable.

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
