# Peer review of "GABA Application Enhances Drought Stress Tolerance in Wheat Seedlings (Triticum aestivum L.)"

_plants, 2023, doi:10.3390/plants12132495_

Round 1
Reviewer 1 Report
The paper "Effects of GABA on the phenolic acids accumulation and antioxidant system in wheat (Triticum aestivum L.) seedlings under drought stress" is well written, however, some corrections and additional clarifications are needed. The comments are in the attachment.

Author Response
We are very grateful to you for the comments and advices on our manuscript (ID is plants-2404012). Those comments are all valuable and very helpful for revising and improving our manuscript, as well as the important guiding significance to our researches. We have studied the comments carefully and have made correction which we hope meet with approval. We are willing to do any further revision if necessary. The main corrections in the paper and the responds are in the attachment.

Reviewer 2 Report
"The article titled 'Effects of GABA on the Phenolic Acid Accumulation and Antioxidant System in Wheat (Triticum aestivum L.) Seedlings under Drought Stress' by Zhao et al. appears intriguing. To enhance its clarity, I suggest condensing the title by focusing on the effect of GABA under drought stress, rather than explicitly mentioning phenolic acid accumulation and antioxidant system.
In the introduction section, it would be beneficial for the authors to provide a more comprehensive explanation of the GABA shunt pathway in wheat and its connection to abiotic stress, such as high salinity and drought. While numerous studies have explored GABA's role under salinity in wheat, there is a scarcity of research on its role under drought. Therefore, the authors should emphasize the importance of investigating GABA's role specifically in the context of stress.
Figure 1: It would be valuable to include information about the number of replications used in this experiment to ensure statistical rigor and reproducibility.
Table: It is surprising to note that phenolic acid was not detected in wheat seedlings for many of the treatments. This observation raises interesting questions and could be further discussed.
Discussion section: The authors have not addressed the specific role of GABA in explaining the observed metabolic changes. Additionally, it would be beneficial to explore the differences between GABA's role under salinity and drought conditions, providing a comparative analysis of its effects in these distinct stress environments."
minor edited and polishing required
Author Response
We are very grateful to you for the comments and advices on our manuscript (ID is plants-2404012). Those comments are all valuable and very helpful for revising and improving our manuscript, as well as the important guiding significance to our researches. We have studied the comments carefully and have made correction which we hope meet with approval. We are willing to do any further revision if necessary. The main corrections and the responds are in the attachment.

Round 2
Reviewer 1 Report
The manuscript is much better after corrections, however it is very difficult to assess the transparency of the tables since word template was not used for the manuscript. Regardless of the above, I still think Table 1 is too large and unreadable. I suggest it be transferred to the supplement, and only the ANOVA results for the analyzed properties should be used.
Author Response
Thanks so much for your general support for our manuscript, we have make revisions following your suggestions. Since the data in Table 1 is very important and closely related to the research content, we did not move it to the attached table. However, according to your suggestion, we changed the way of presentation to a graph, which looks very intuitive. We are willing to do any further revision if necessary.

Reviewer 2 Report
The authors have addressed all the concerns and revised the MS as expected. I recommend acceptance of the current version of the MS.
Author Response
Thanks so much for your general support for our manuscript.